METHODS AND PROTOCOLS

# Metagenomic Sequencing for Identification of *Xylella fastidiosa* from Leaf Samples

Verónica Román-Reyna,[a,b] Enora Dupas,[c,d] Sophie Cesbron,[c] Guido Marchi,[e] Sara Campigli,[e] Mary Ann Hansen,[f] Elizabeth Bush,[f] Melanie Prarat,[g] Katherine Shiplett,[g] Melanie L. Lewis Ivey,[h] Joy Pierzynski,[i] Sally A. Miller,[b,h] Francesca Peduto Hand,[a] Marie-Agnes Jacques,[c] Jonathan M. Jacobs[a,b]

aDepartment of Plant Pathology, The Ohio State University, Columbus, Ohio, USA

bInfectious Disease Institute, The Ohio State University, Columbus, Ohio, USA

cUniversity of Angers, Institut Agro, INRAE, IRHS, SFR QUASAV, Angers, France

dFrench Agency for Food, Environmental, and Occupational Health & Safety, Plant Health Laboratory, Angers, France

eDepartment of Agriculture, Food, Environment, and Forestry, University of Florence, Florence, Italy

fSchool of Plant and Environmental Sciences, Virginia Tech, Blacksburg, Virginia, USA

gAnimal Disease Diagnostic Laboratory, Ohio Department of Agriculture, Reynoldsburg, Ohio, USA

hDepartment of Plant Pathology, The Ohio State University, Wooster, Ohio, USA

iC. Wayne Ellett Plant and Pest Diagnostic Clinic, Department of Plant Pathology, The Ohio State University, Reynoldsburg, Ohio, USA

**ABSTRACT** *Xylella fastidiosa* (*Xf*) is a globally distributed plant-pathogenic bacterium. The primary control strategy for *Xf* diseases is eradicating infected plants; therefore, timely and accurate detection is necessary to prevent crop losses and further pathogen dispersal. Conventional *Xf* diagnostics primarily relies on quantitative PCR (qPCR) assays. However, these methods do not consider new or emerging variants due to pathogen genetic recombination and sensitivity limitations. We developed and tested a metagenomics pipeline using in-house short-read sequencing as a complementary approach for affordable, fast, and highly accurate *Xf* detection. We used metagenomics to identify *Xf* to the strain level in single- and mixed-infected plant samples at concentrations as low as 1 pg of bacterial DNA per gram of tissue. We also tested naturally infected samples from various plant species originating from Europe and the United States. We identified *Xf* subspecies in samples previously considered inconclusive with real-time PCR (quantification cycle [$C_q$], >35). Overall, we showed the versatility of the pipeline by using different plant hosts and DNA extraction methods. Our pipeline provides taxonomic and functional information for *Xf* diagnostics without extensive knowledge of the disease. This pipeline demonstrates that metagenomics can be used for early detection of *Xf* and incorporated as a tool to inform disease management strategies.

**IMPORTANCE** Destructive *Xylella fastidiosa* (*Xf*) outbreaks in Europe highlight this pathogen's capacity to expand its host range and geographical distribution. The current disease diagnostic approaches are limited by a multiple-step process, biases to known sequences, and detection limits. We developed a low-cost, user-friendly metagenomic sequencing tool for *Xf* detection. In less than 3 days, we were able to identify *Xf* subspecies and strains in field-collected samples. Overall, our pipeline is a diagnostics tool that could be easily extended to other plant-pathogen interactions and implemented for emerging plant threat surveillance.

**KEYWORDS** *Xylella fastidiosa*, metagenomics, diagnostics, short-read sequencing

Address correspondence to Jonathan M. Jacobs, jacobs.1080@osu.edu.

**X**ylella fastidiosa (*Xf*) is a globally distributed insect-transmitted plant-pathogenic bacterium causing diseases on a large host range. To date, 595 plant species belonging to 85 botanical families have been reported as *Xf* hosts (1), some of which are of major socioeconomic interest, such as grapevine, olive, citrus, coffee, and almond (2). *Xf* colonizes the

mSystems®

xylem vessels of plants, where it forms biofilms (3) that, together with tyloses and gums produced by the plant in response to the infection (4), limit water translocation. Infected hosts display symptoms of leaf scorches and plant dieback finally followed by plant death (3).

*Xf* was first described in and limited to the Americas but recently emerged in Europe, highlighting the pathogen's capacity to expand its host range and geographical distribution (2, 5). The pathogen was reported in Italy in 2013, where is currently devastating Apulian olive production, then detected in France in 2015, Spain in 2016 and Portugal in 2018, on both cultivated and spontaneous Mediterranean plant species (2). The primary control strategy for diseases caused by *Xf* includes eradication of infected hosts via early detection. Therefore, developing methods with the lowest limit of detection are critical to prevent major losses for growers and future pathogen dispersal.

The diagnostics of diseases caused by fastidious pathogens such as *Xf* are difficult. This difficulty is increased, as infected plants may remain asymptomatic for very long periods of time, which is associated with low bacterial concentrations and an irregular distribution of the pathogens in the plants (6). It is of major interest to develop reliable and highly sensitive tools for detection and detailed identification that can be used directly on plant extracts. Current standards for *Xf* diagnostics primarily rely on quantitative real-time PCR (qPCR) assays to detect and sometimes identify the bacterium (7–12), followed by the amplification and sequencing of two, for subspecies identification, to seven housekeeping genes (*cysG*, *gltT*, *holC*, *leuA*, *malF*, *nuoL*, and *petC*) for sequence type (ST) determination and phylogeny reconstruction (2) (Fig. 1A). Five subspecies are proposed in *X. fastidiosa*, i.e., *fastidiosa*, *multiplex*, *pauca*, *morus*, and *sandyi* (13–15). However, whole-genome analyses revealed similarities of the subspecies *fastidiosa*, *morus*, and *sandyi*, which cluster into one clade. Moreover, genome analysis indicated a high frequency of horizontal gene transfer and recombination among *Xf* subspecies (14–16).

Plant samples infected by more than one *Xf* strain belonging to several subspecies are not uncommon and are not easy to detect (17, 18). Nevertheless, current methods do not consider new or emerging variants resulting from pathogen genetic recombination (14). For example, qPCR with high quantification cycle ($C_q$) values (>35) are considered inconclusive (2), making decisions about disease control difficult. A complementary tool for diagnostics is the use of next-generation sequencing (19) (Fig. 1). Because this approach can be directly used on plant extracts, it is not biased toward known sequences and provides more information about the pathogen genome, such as virulence traits. Metagenomics, the study of genetic material from environmental samples, beyond whole-genome sequencing, allows for the detection of strains from several subspecies and ST at the same time from the host (20). Recently, the use of long-read sequencing as diagnostic tool identified *Xf* subspecies and ST from infected samples (12, 21).

In this study, we developed and tested a metagenomics pipeline using in-house short-read sequencing as a complementary approach for affordable and accurate *Xf* detection. We were able to use metagenomics to identify *Xf* to the strain level in single- and mixed-infected plant samples, at concentrations as low as 1 pg of bacterial DNA per gram of tissue. In addition, we tested naturally infected field samples from Europe and the United States. We identified *Xf* subspecies in samples with $C_q$ values equal to and greater than 37, which is beyond the threshold of detection for the standard and certified qPCR methods (2). Overall, we developed a robust diagnostics pipeline that could be easily extended to other pathogens and implemented for surveillance of emerging agricultural threats.

## RESULTS

**Metagenomics for the diagnostics pipeline.** We developed and tested a metagenomics pipeline for *Xf* detection and subspecies identification (Fig. 1). We tested this pipeline based on three types of DNA samples—from bacterial colonies in culture, spiked plant samples, and naturally infected plant samples (Fig. 1B). To recover and identify *Xf* subspecies and compare them to the already sequenced genomes, we developed a pipeline that uses six different tools and custom-made databases (22) (Fig. 1C). The pipeline recovers *Xf* reads with the software Kraken 2 and a custom-made database (22). The database has

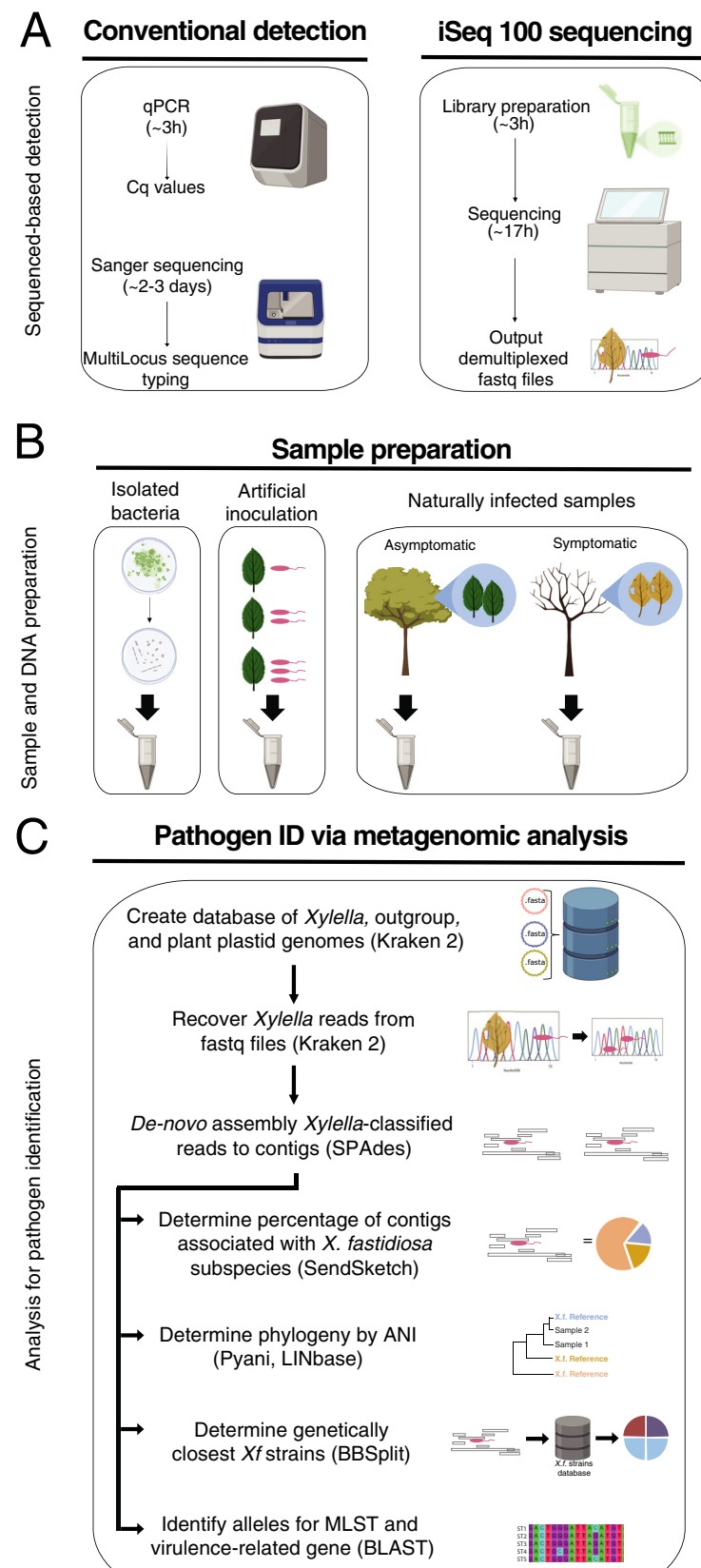

**FIG 1** Metagenomics for the diagnostic pipeline. (A) Sequenced-based detection. Two approaches were used for *Xf* detection—conventional detection and iSeq 100 sequencing. For conventional

user-specified genomes for *Xf* read identification. The user-specified genomes belonged to *Xylella* (*n* = 81), *Xanthomonas* sp. (*n* = 10), *Escherichia coli* (*n* = 1), and several plant sequences from NCBI (Table S1). The custom-made database had plant sequences to avoid false-positive results because we found that some NCBI *Xf* genomes contained plant genomic DNA sequences. The plant DNA sequence hits had 100% identity to plant 18S or chloroplast reads. We could not remove all plant reads from the 81 NCBI *Xf* genomes. Therefore, the plant reads in the database serve as a filter to ensure that plant reads were not misidentified as *Xf* reads.

After Kraken 2, the pipeline *de novo* assembled the recovered *Xf* reads into contigs with the program SPAdes (23). The pipeline used the *Xf* contigs for four different analyses: (i) subspecies identification, (ii) phylogeny reconstruction, (iii) identification of the already sequenced genetically closest strains, and (iv) alleles for multilocus sequence type (MLST) profile and virulence-related genes determination. The pipeline used *Xf* contigs and the tool SendSketch from the BBMap software to identify subspecies. Then it used Pyani and LINbase software to reconstruct phylogeny by average nucleotide identity (ANI) (24, 25). Next, it assigned *Xf* strains to each *Xf* contig to identify the closest strains with the tool BBSplit from the BBMap software. Finally, to identify specific genes or alleles, the pipeline used local BLAST search with two types of subjects; (i) one subject was the reported MLST allele genes, and (ii) the other subject was the protein sequences from genes associated with virulence.

**Xf identification from *in silico* prepared samples.** To test the pipeline sensitivity, we used *in silico* samples with target (e.g., *Xf*) and nontarget DNAs (e.g., nonhost plant). The samples included variable amounts of nonhost barley (*Hordeum vulgare*) sequenced reads *in silico* spiked with *Xylella fastidiosa* subsp. *fastidiosa* (*Xff*) CFBP 7970 reads. We obtained a strong linear correlation between the Kraken 2 results and the proportion of spiked *Xf* sequence reads (percentage of *Xylella* reads spiked in sample, $y = 103.21x - 0.0127$; $R^2 = 1$) (Fig. S1A). We recovered *Xf* reads and assembled them as contigs using SPAdes. With the *Xf* contigs, we performed BLAST analysis to identify MLST alleles and virulence genes. We were able to identify one to four MLST-related genes for samples spiked with 0.5 to 2.4% *Xf* reads (Table S3). This result indicated that we cannot capture the full MLST gene set for ST identification with less than 2.4% *Xf* reads (Table S3). We calculated, for all samples, the percentage of gene similarity to the virulence-related genes (Table S4). The percentage of gene similarity increased with the higher number of spiked *Xf* reads. Samples with a lower number of *Xf* reads had a low genome coverage to recover and analyze complete gene sequences (Tables S4 and S6).

We then identified *Xf* subspecies using the *Xf* contigs. Since the *in silico* samples only identified *Xff* reads, we expected that SendSketch assigned all contigs to *Xff*. However, we found that 9 to 15% of *Xff* contigs were instead assigned to *Xfm*. Based on these results, we did two additional analyses to determine the best approach to analyze the *Xf* subspecies composition. For the first analysis, we hypothesized that complete assembled genomes

**FIG 1** Legend (Continued)

detection, samples were analyzed using qPCR assays, Harper's test or tetraplex Dupas's test, and MLST involving Sanger sequencing of seven housekeeping genes. iSeq 100 libraries were prepared according to the manufacturer's instructions. After 17 h of sequencing, demultiplexed samples were recovered from the machine and used for subsequent analysis. (B) Sample preparation. The samples used for the pipeline were DNA-extracted from bacterial strains in culture, spiked plant material, and naturally infected samples. (C) Pathogen identification via metagenomic analysis. Demultiplexed fastq reads from all samples were then used for metagenomic analysis. We created a database to recover *Xf* reads using Kraken 2. The database contained *Xylella*, *Xanthomonas*, and *Escherichia coli* genomes. We also added plant plastid genomes to remove false-positive results. *Xf* reads were recovered from the fastq files. The *Xf* recovered reads were *de novo* assembled to obtain *Xf* contigs, using SPAdes. The *Xf* contigs were used in four different analyses, subspecies identification, phylogeny reconstruction, identification of the genetically closest strain with a sequenced genome, and alleles from specific genes. To determine subspecies, we used the tool SendSketch. To reconstruct phylogeny, we calculated ANI using Pyani and the website tool LINbase (https://linbase.org/). To determine the genetically closest known *Xf* strain, we detected the number of hits to each *Xf* strain using BBSplit. To identify specific gene alleles, we calculated the percentage of identity to the seven MLST genes (*cysG*, *gltT*, *holC*, *leuA*, *malF*, *nuoL*, *petC*) and the percentage of similarity to 17 virulence-related proteins using local BLAST+. Graphics were created with BioRender.

would reduce the percentage of reads assigned to other subspecies. To test this, we created a smaller Kraken2 database with 30 genomes instead of 81. These 30 *Xf* genomes had complete assemblies. We recovered 1% to 2% fewer *Xf* reads with the new database, and the subspecies distribution remained the same (data not shown). The results indicated that *Xf* subspecies classification is not related to the level of genome assembly. Therefore, the original Kraken2 database with 81 NCBI *Xf* genomes was retained for all further analyses.

For the second analysis, we manually assessed the SendSketch sensitivity to mixed infections with new *in-silico* samples. The new samples included a set amount of barley reads *in silico* spiked with variable amounts of *Xff* CFBP 7970 and *X. fastidiosa* subsp. *multiplex* (*Xfm*) CFBP8418 reads (Table S2). For these new *in silico* samples, we recovered *Xf* reads with Kraken 2, assembled the reads as contigs, and ran SendSketch to identify subspecies. When using BLAST, a certain number of *Xf* contigs mapped equally (100% identity) to *Xff* and *Xfm* (*Xf* core contigs). We observed that *Xf* core contigs are directly proportional to the total of *Xf* recovered reads and samples with a higher *Xff* to *Xfm* spiked read ratio (Table S2, Fig. S1B and C). Moreover, the tool SendSketch randomly assigned the subspecies to *Xf* contigs with 100% identity to *Xff* and *Xfm*. Consequently, we developed a manual correction to separate single from mixed infections. We only used samples with either *Xfm* or *Xff*; *X. fastidiosa* subsp. *pauca* (*Xfp*) was not part of the analysis. The correction consists of calculating the logarithm of the *Xfm*:*Xff* contig ratio. Corrected log ratios from 0.081 to 0.4 are considered a mixed infection. Log ratios below $-0.012$ are identified as single *Xff*, and those higher than 0.43 are *Xfm* single infection (Fig. S1B and C).

To evaluate the pipeline with samples free of *Xf*, we used extracted DNAs of two healthy plant samples and a non-*Xylella* control (i.e., barley leaves infiltrated with *Xanthomonas*). For the artificially inoculated barley samples, Kraken 2 software recovered 20 to 30% of the total reads as *Xanthomonas*. For all four *Xf*-free samples, Kraken 2 recovered 6 to 19 *Xf* reads (Table S2). All these *Xf* reads corresponded to plant reads based on the BLAST Web tool from NCBI. Based on these results, the pipeline considers a sample *Xf*-free when it cannot recover more than 19 *Xf* reads (Fig. S2).

**Xf identification from isolated bacteria.** To test the capacity of iSeq 100 sequencing, we used six genomic DNAs (gDNAs) from isolated bacteria and two known *Xf* gDNAs as controls. The six *Xf* gDNAs were isolated from Italian field samples (See Materials and Methods). The two control *Xf* gDNA samples were *Xff* CFBP 7970 (CFBP 7970 iSeq100) and *Xfm* CFBP 8418 (CFBP 8418 iSeq100). All eight gDNA samples were sequenced with the iSeq 100 system. For all eight samples, Kraken 2 recovered 99% of the total reads as *Xf* reads (Table S2). We assembled the *Xf* reads as contigs and classified them into subspecies. For CFBP 7970 and CFBP 8418, for which a genome was already available, 54 to 78% of the contigs corresponded to *Xf* core contigs, 44 to 65% to their respective subspecies, and 2% to the closest subspecies. On average, within the six Italian samples, 20% of the contigs corresponded to *Xf* core contigs, 25% to *Xfm*, and 2% to *Xff*.

The ANI values were consistent with the *Xf* contig abundance (Fig. 2, Table S2). All six Italian samples and CFBP 8418 iSeq100 had 99 to 100% identity to *Xfm* and less than 97% identity to *Xff* and *Xfp*. The control sample, CFBP 7970 iseq100, had 100% identity to *Xff* and less than 98% identity to *Xfm* and *Xfp* (Fig. 2, Table S2).

For strain identification, we used the program BBSplit and the Harvest suite. For each sample, we selected the top three closest strains based on the program BBSplit output. Then, we used these closest strains and the sample to compare the number of single nucleotide polymorphisms (SNPs) with the Harvest suite (26). For the sample CFBP 8418 iSeq100, the closest strain with 30 SNPs was *Xfm* CFBP 8418 (Tables S3 and S5). For the sample CFBP 7970 iSeq100, the closest strain with fewer SNPs was *Xff* CFBP 7970. The six Italian samples had the same three closest *Xfm* strains, TOS5, TOS4, and TOS14. All six samples had fewer SNPs than the strain *Xfm* TOS4. The three TOS strains and the Italian samples were isolated from the outbreak area of Monte Argentario, Tuscany, Italy (27).

We performed BLAST analysis to identify MLST alleles and virulence genes for all eight isolated bacteria with the assembled *Xf* contigs. For virulence genes, the sample CFBP 7970 iSeq100 had 100% similarity to all *Xff* virulence genes except for rpfE (96.4%)

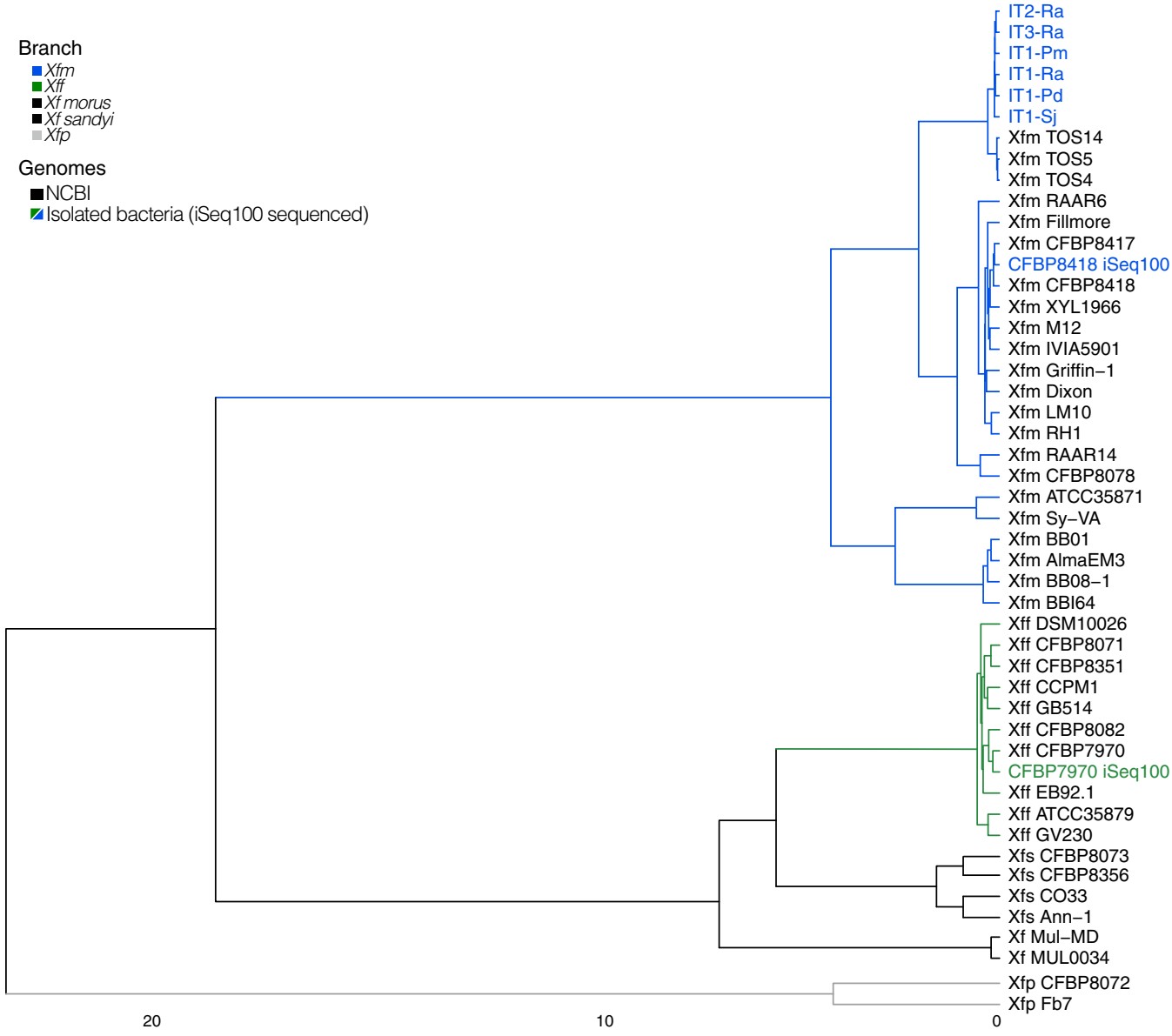

**FIG 2** Phylogenetic reconstruction of isolated bacteria used in this study. The cluster analysis is based on average nucleotide identity values from Pyani. Branch colors indicate different *Xf* subspecies—*Xff* (green), *Xfm* (blue), *Xfp* (gray), *Xf* subspecies *morus* and *sandyi* (black). The sequenced gDNA from isolated bacteria are indicated in blue or green. The *Xf* genomes obtained from NCBI are indicated in black. The cluster was plotted using ComplexHeatmap R package.

(Table S4). The sample CFBP 8418 iSeq100 had 100% similarity to all *Xfm* virulence genes except for pilB (99.8%). The six Italian samples had the same similarity percentages for all *Xfm* virulence genes except for hemagglutinin (95.7 to 100%). We were able to identify all virulence genes and to complete the allelic profiles for ST identification (Table S3). As expected, the ST identified for the sample CFBP 7970 iSeq100 was ST2, and the one for CFBP 8418 iSeq100 was ST6. The six Italian samples had the same ST87 number.

**Xf identification from spiked plant samples.** We tested the pipeline with DNA extracted from grapevine petioles and midribs artificially inoculated with known bacterial concentrations of the strain *Xff* CFBP 7970, *Xfm* CFBP 8418, or an equal mix of both strains. Kraken 2 output recovered 0.01 to 74.8% of the total sequences as *Xf* reads (Table S2). The percentage of recovered *Xf* reads had a positive correlation with $\log_{10}$ CFU values ($R^2 = 0.9876$) (Fig. 3A) and $C_q$ values ($R^2 = 0.9141$) (data not shown). The pipeline detected *Xf* with the lowest bacterial concentration tested in this study ($1 \times 10^4$ CFU/ml), equivalent to a $C_q$ value of 28.85 and 1.62 pg·$\mu$l$^{-1}$.

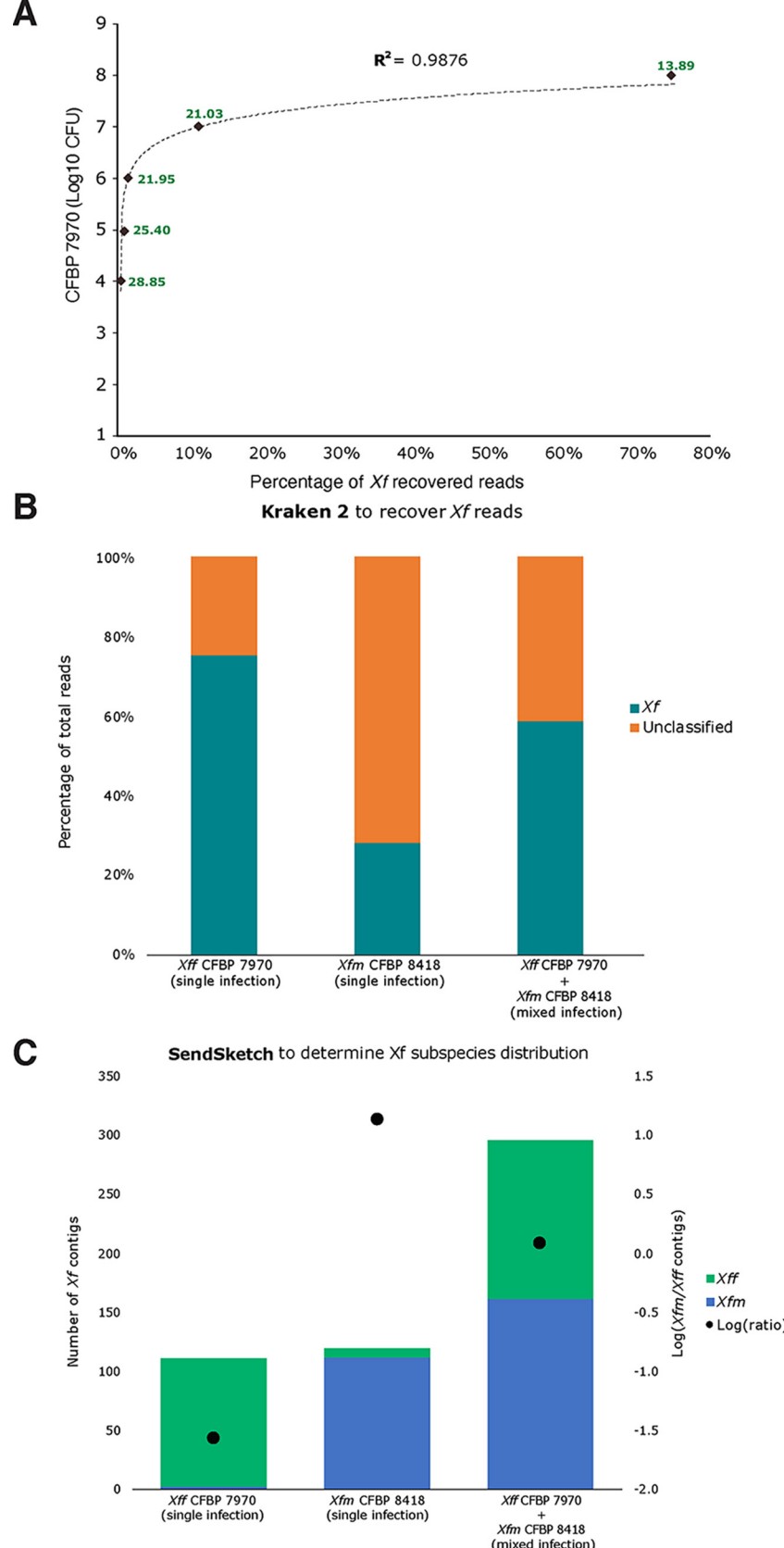

**FIG 3** Spiked samples with different dilutions and mixed samples. (A) Scatterplot comparing the log$_{10}$ CFU with the percentage of *Xff* CFBP7970 recovered reads from the total read number. The dotted

After *Xf* contig assembly, we were able to identify the subspecies for all samples, and the log ratio separated single from mixed infections (Fig. 3B and C). The log ratio for *Xff* single infections varied between −1.56 and −0.23. The log-ratio for *Xfm* single infection was 1.15, while that for mixed-strain infection was 0.08. The ANI values confirmed the *Xff* and *Xfm* subspecies for single-infected samples (Fig. S3, Table S2). The mixed sample (*Xff* CFBP 7970 + *Xfm* CFBP 8418) had a higher ANI value for *Xfm*. This result was consistent with a higher number of *Xfm* contigs for the mixed-strain sample (Fig. 3C, Table S2).

Based on BBSplit results, the genetically closest strains sequenced for most of the *Xff* single infections was CFBP 7970, followed by ATCC 35879, GV230, and TPD4 (Table S5). For *Xfm* single infection, the genetically closest *Xfm* strains were Dixon and CFBP 8418. For the mixed infection, 90% of *Xf* contigs were assigned to *Xfm* Dixon and 6% to *Xff* strains.

With the assembled *Xf* contigs, we performed BLAST analysis to identify MLST alleles and virulence genes. We identified the ST number for three of the seven artificially inoculated samples. The mixed-infected sample of grapevine inoculated with the strain CFBP 8418 was identified as ST6 (Table S4). The grapevine sample inoculated with CFBP 7970 ($10^8$ CFU/ml) was ST2. We could not assign an ST to the grapevine sample inoculated with the strain CFBP 7970 ($10^7$ CFU/ml) because we only identified six of the seven MLST alleles. In contrast to MLST analysis, we detected at least two virulence genes per sample (Table S4). The sample with the lowest CFU values, CFBP 7970 ($10^4$ CFU/ml), had 42% similarity to *Xff* hemagglutinin and 41% similarity to *Xfm* pilQ. For the remaining *Xff* single-infected samples, the percentage of similarity to a single subspecies increased with the higher CFU number, which is also associated with higher genome coverage (Table S6). For the *Xfm* single infection, all the virulence genes had 100% similarity to *Xfm*. For the mixed infection, all the virulence genes had 100% similarity to *Xfm* and, on average, 98% to *Xff*.

***Xf* identification from field-collected samples.** Finally, we tested the iSeq 100 sequencing capacity with European and American field samples (Table S2). We used 24 samples with $C_q$ values ranging from 21 to 40 based on Harper's qPCR assay. We used three samples that were negative based on the same qPCR assay. The DNAs from the 27 samples were extracted from six different hosts—*Olea europaea*, *Polygala myrtifolia* (France and Italy), *Quercus ilex*, *Spartium junceum*, *Rhamnus alaternus*, and *Vitis vinifera*.

Kraken 2 recovered 0.004 to 1.43% of the total reads as *Xf* (Table S2). We assembled the *Xf* reads into 1 to 2,896 contigs. We found all samples had at least one contig with at least 400 bp. The limit of *Xf* detection with Harper and tetraplex qPCR corresponded to 30 to 37 $C_q$ values (17). Therefore, we evaluated 16 samples that either had fewer than 30 *Xf* contigs, were classified as *Xf*-negative, or had $C_q$ higher than 30 (2). We used each *Xf* contig from the 16 samples as the query for a nucleotide BLAST search using the Web tool from NCBI. Of the 16 samples, 11 gave 100% identity to *Xf* genomes. Hence, these 11 samples were considered *Xf* positive. All contigs from the other five samples, FR1-Pm, FR1-Oe, IT6-Sj, IT11-Sj, and US1-Vv, had 100% identity to chloroplast and 18S plant sequences but none to *Xf*. Therefore, these five samples were considered *Xf* negative (Fig. 4). With our pipeline, we were able to detect *Xf* in samples considered inconclusive by qPCR according to Harper's (2).

We then used the contigs from the 22 *Xf* positive samples for subspecies classification. Overall, the samples had 50% to 100% of contigs classified as *Xf* core contigs (Fig. 4). Three French samples (FR2-Qi, FR2-Pm, FR4-Oe) and five Italian samples (IT5-Sj, IT7-Sj, IT8-Sj, IT9-Sj, IT6-Ra) had 1 to 6 contigs assigned as *Xfm*. The French sample FR3-Oe, seven Italian samples (IT2-Sj, IT3-Sj, IT4-Sj, IT12-Sj, IT4-Ra, IT2-Pm, IT3-Pm), and the U.S. sample, US2-Vv, were *Xfm* single-infected based on the manual log correction (log ratio > 0.43) (Table S2). The sample

**FIG 3** Legend (Continued)

line indicates a logarithmic trendline, $y = 0.4222\ln(x) + 7.9511$; $R^2 = 9,876$. The green numbers indicate the $C_q$ values for each sample. (B) Percentage of *Xf* recovered reads by Kraken 2 from single-infected (*Xff*, 1e8; *Xfm*, 1e7 CFU) and mixed-infected (1e7 CFU) samples. Teal bars indicate *Xf* reads, and orange bars indicate unclassified reads. Unclassified reads show no similarity to *Xf* reads, such as plant or other microorganisms reads. (C) Proportion of *Xf* subspecies from total *Xf* contigs in single and mixed infections. The black dots indicated the log ratio as a manual correction to detect single and mixed infection. *Xfm* and *Xff* are indicated in blue and green, respectively.

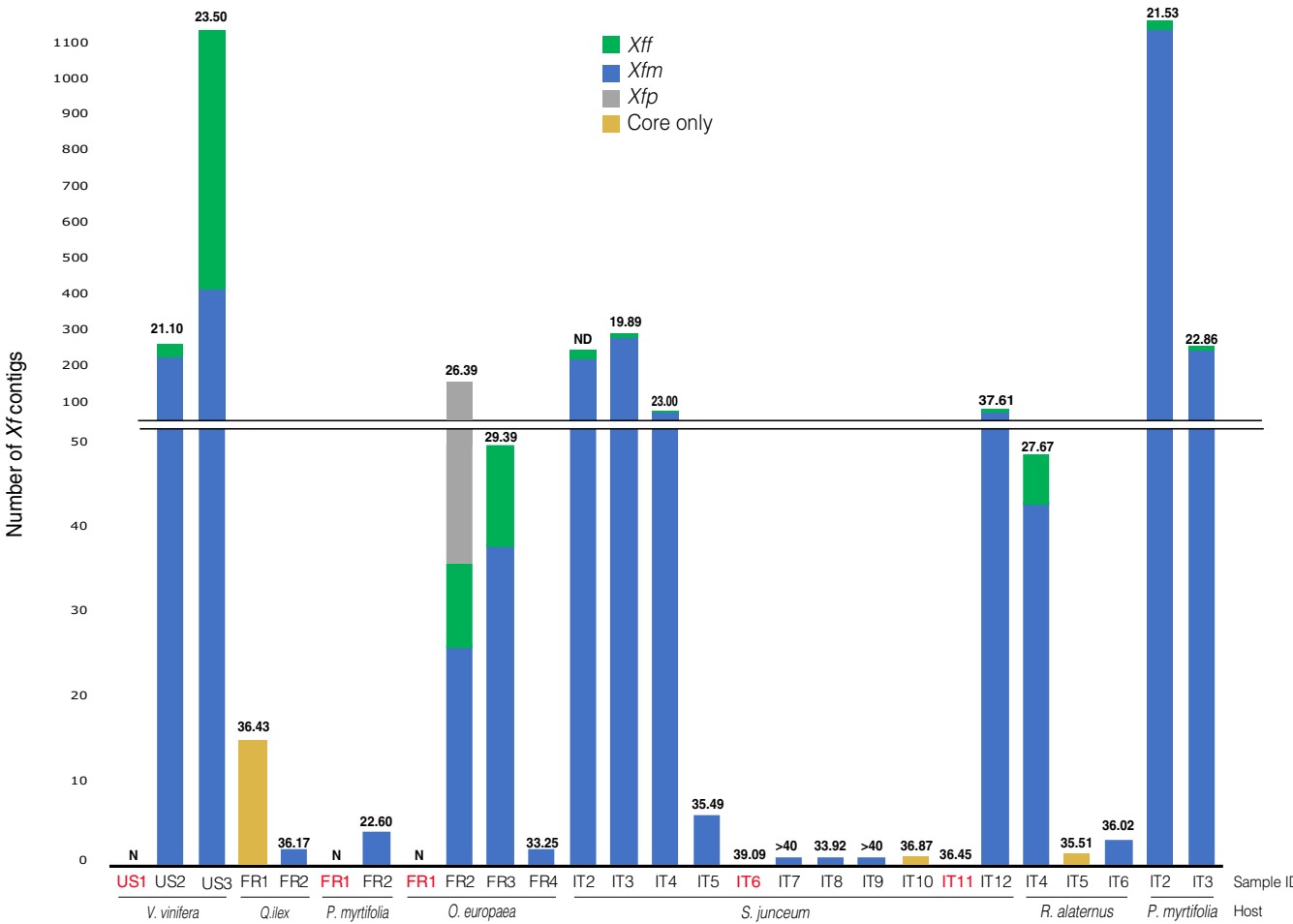

**FIG 4** *Xf* from field-collected samples and read mapping to subspecies from database. Stacked columns indicate the number of *Xf* contigs with 100% identify to each *Xf* subspecies. Samples indicated in red were *Xf*-negative with our pipeline. *Xff* represents the sum *of Xf* subsp. *fastidiosa*, *Xf* subsp. *morus*, and *Xf* subsp. *sandyi*; *Xfm*, *Xf* subsp. *multiplex*; *Xfp*, *Xf* subsp. *pauca*. "Only core" indicates samples that only have *Xf* core contigs. The $C_q$ values are indicated on the top of each bar. ND, not determined; N, negative for *Xf* based on qPCR. Sample code and hosts are indicated on the x axis. Each country of origin is indicated in the sample ID—France (FR), Italy (IT), and United States (US)—along with the host from which they were isolated.

FR2-Oe had 17% of contigs assigned to *Xfp*, and the sample US3-Vv was *Xff* single-infected (log ratio < 0.08).

Then, we determined the ANI values and *Xf* strain compositions for the 22 *Xf*-positive samples. Both results were consistent with the subspecies identification. The Italian samples had 99 to 100% ANI to *Xfm*. Five French samples (FR1-Qi, FR2-Pm, FR3-Oe, FR4-Oe, FR2-Qi) had 99 to 100% ANI to *Xfm*, and FR2-Oe had 98% ANI to *Xfp* (Fig. 5A, Table S2). The sample US2-Vv had 99% ANI to *Xfm*, and US3-Vv had 100% to *Xff* (Fig. 5A). For strain distribution, all the French and U.S. and three Italian samples (IT2-Sj, IT4-Sj, IT5-Sj) had more than 40% *Xf* contigs had 100% identity to one strain (Table S5, Fig. 5B). Except for IT2-Sj, the Italian samples had most of the contigs assigned to the three strains TOS4, TOS5, and TOS14. The sample IT2-Sj had more reads assigned to *Xfm* RAAR14. Overall, the tools SendSketch, Pyani, and Bbsplit validated the qPCR subspecies results for field samples.

We performed BLAST analysis to identify MLST alleles and virulence genes for the 22 infected samples. For 16 of 22 *Xf*-positive samples, we found the percentage of gene similarity to be 26 to 100% for at least one virulence gene (Table S4). Distinct from single-gene analysis, we only identified some MLST-related alleles for four samples (US3-Vv, FR2-Oe, IT2-Pm, IT3-Pm); consequently, we could not identify the ST number (Table S3).

To compare some of our results with a high-performance, deep-sequencing Illumina platform as a control, we selected nine samples for resequencing with the MiSeq platform

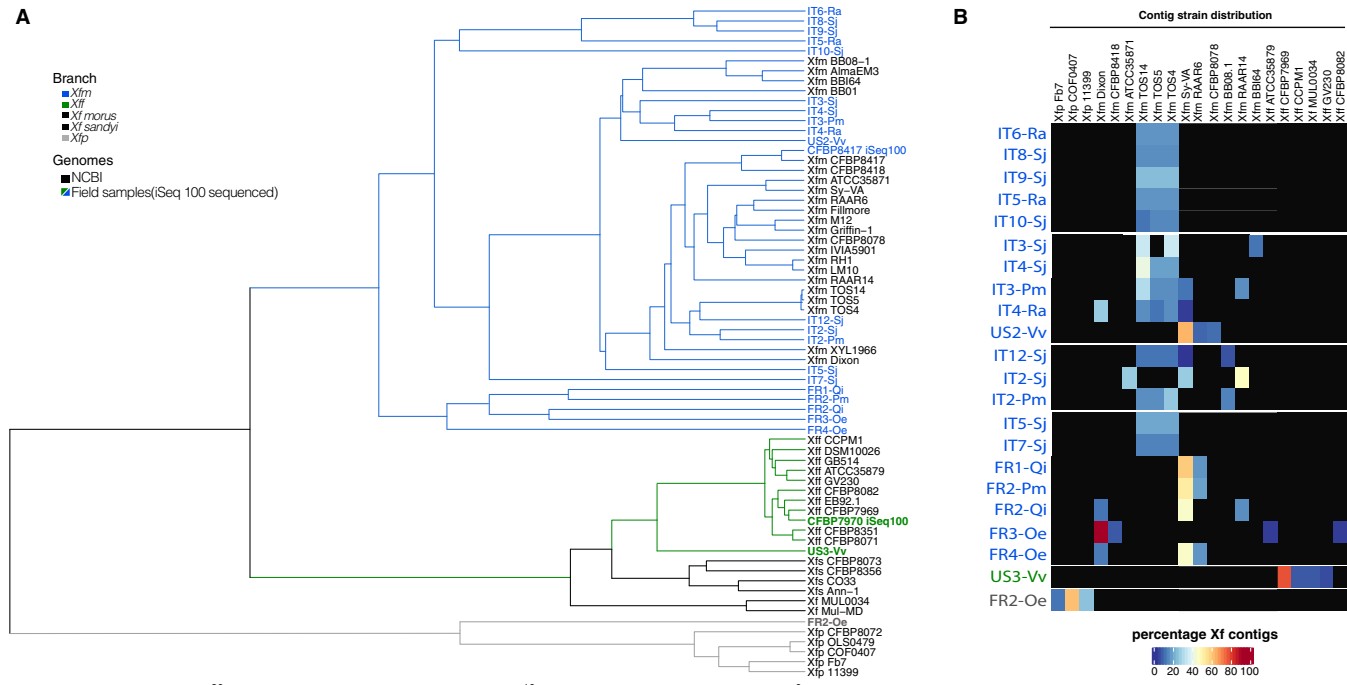

**FIG 5** Metagenomics analyses of field sample identifies of bacterial subspecies. (A) The dendrogram indicates the distance and cluster analysis based on ANI values using NCBI whole genomes and assembled *Xf* contigs. Branch colors represent each *Xf* subspecies. Blue, gray, and green names indicate iSeq 100-sequenced samples. (B) The heatmap shows the percentage of unique contigs assigned to each *Xf* strain. The cluster and strain distribution were plotted using ComplexHeatmap R package.

using the same iSeq 100 libraries from this study. The nine samples were IT3-Pm, IT5-Sj, FR2-Oe, FR3-Oe, US1-Vv, US2-Vv, FR1-Pm, FR2-Pm, and IT9-Sj (Table S2, Fig. S4). We analyzed the MiSeq sequences with our pipeline and recovered 0.005 to 0.792% of total reads as *Xf* reads with Kraken 2. We assembled the *Xf* reads into contigs and manually assessed all samples with less than 30 *Xf* contigs. The NCBI blastn analysis indicated that the samples US1-Vv and FR1-Pm had contigs with 100% identity to plant reads; therefore, we confirmed they were *Xf*-negative samples. The other seven samples were considered *Xf*-positive. We followed the pipeline to identify and determine subspecies, phylogeny, genetically closest sequenced genome strains, MLST profile, and virulence-related genes. The results for subspecies and phylogeny identification were the same between MiSeq and iSeq 100 sequencers (Fig. S4), but there were some differences for the other three analysis results (Table S2). For the genetically closest sequenced genome strain analysis, all samples gave the same strain distribution as iSeq 100 results, except FR2-Oe, which showed *Xfp* OLS0478 instead of *Xfp* COF0407 as the most abundant strain. These two *Xfp* strains are phylogenetically close. For MLST analysis, we identified four more alleles for the sample IT3-Pm with the MiSeq platform than with iSeq 100 (Table S3), while we only detected two alleles in the sample US2-Vv sequenced with Miseq. We were only able to detect MLST alleles for the sample FR2-Oe with the iSeq 100 platform. We were able to calculate the percentage of gene similarity for more virulence genes with the MiSeq platform than with the iSeq 100 platform. The variation between Illumina platforms was not consistent.

## DISCUSSION

In this study, we developed a user-friendly metagenomic pipeline to identify and determine *Xylella fastidiosa* subspecies from field-collected samples without the need for pathogen isolation. We demonstrated the flexibility of the pipeline by using seven different plant hosts and three DNA extraction methods. We recovered and assembled *Xf* reads into contigs from total DNA samples. We used percentage of similarity to a single subspecies to identify *Xf* subspecies and validated the results through phylogeny and

strain proximity. Finally, we examined potential virulence-related genes among all sequenced samples.

To recover *Xf* reads from field samples with the tool Kraken 2, we used *Xf* genomes available on NCBI. We found plant plastid reads in all genomes obtained from pure *Xf* cultures, except for *Xff* CFBP7970. We decided to add plant reads in the database to filter out potential plant contamination. We still recovered *Xf* reads for some plant samples reported as *Xf*-negative. Therefore, we manually examined the contigs of samples with fewer than 30 *Xf* contigs to determine if they have a low *Xf* concentration or are negative samples. More than the $C_q$ values, the contig evaluation will be necessary to determine if a sample is truly *Xf*-negative.

We observed that the Kraken 2 tool not only recovered *Xf* reads but also identified *Xf* subspecies. We decided not to use Kraken2 to identify subspecies because we found that the recovery is affected by incomplete NCBI *Xf* genome subspecies information. Kraken 2 uses by default the NCBI taxonomy to classify reads; if the genomes used to build the database do not have subspecies information, it will keep most of the reads at the species level. To improve the subspecies resolution, we decided to use contigs instead of reads and the tool SendSketch (BBMap tool). Contigs or assembled reads increase the coverage and reduce false-positive reads (28). We used SendSketch because it uses the MinHash algorithm to be fast and it takes into account whole genomes and does not use taxonomy.

After we identified samples as *Xf*-positive, we defined *Xf* subspecies. We observed that some samples had mapped contigs to both *Xff* and *Xfm*. These are contigs most likely associated with core sequences, as only 3% of *Xff* and *Xfm* genomes are different. However, it is also possible to have a percentage of annotation error due to sequencing contamination (29, 30). To determine if the samples were single- or mixed-infected, we corrected the results by calculating the log *Xfm*:*Xff* contig ratio (see Materials and Methods). Once we defined the log values for *Xff*, *Xfm*, and *Xfm*+*Xff* (mixed) infections, we validated the presence of single *Xfm* infections in all the tested European samples. Identifying *Xf* subspecies in ornamental and crop plants is essential for the correct application of eradication measures or for plant movements within the European Union territory according to regulation (EU) 2020/1201 (31).

The number of reads generated by the iSeq 100 sequencer highlighted some limitations of the pipeline. For example, the ST were not determined for any field sample because we could not recover the complete sequences of the seven genes. This is probably associated with low genome coverage. The low number of reads also hampers deep SNP diversity, inter-subspecific homologous recombination analyses, and microbiome studies. Other sequencing systems, with higher read output than the iSeq 100, can also be used with this pipeline, as the input is fastq files. With a high number of reads, the pipeline will provide better resolution to recover the MSLT genes and have enough sequencing depth to describe the microbial diversity in the sample. Complete diagnostic analysis should include microbiome diversity analysis to provide the context of microbial interactions during disease.

Some of the diagnostic tools for *Xf* diagnostics are qPCR and Sanger sequencing. These tools require amplification of known *Xf* genome regions but, as is the case of MLST, do not consider new or emerging variants. Moreover, these tools introduce bias due to primer design, have unresolved results with high $C_q$ values, and may take longer since they follow a multistep process. Our pipeline complements these conventional tools by obtaining metagenomic data directly from symptomatic or asymptomatic samples and increasing the detection power. We found some discrepancies between the number of recovered *Xf* reads and $C_q$ values. These differences could be caused by PCR inhibitors or the genomic target region for qPCR that underestimates the bacterial concentration (32, 33). Metagenomics sequencing is becoming a more affordable and faster approach for diagnostics. For example, the whole detection/identification with qPCR and MLST scheme could have an estimated cost of $52 to $54 per sample and takes three to 4 days to detect one to seven genes. With the iSeq 100, it could cost $50 to $70 (when having 12 samples in the same run) and take 2 days but while also allowing a complete genomic analysis of the plant and its pathogenic

and commensal microbiota. Metagenomics for pathogen diagnostics will serve as a model for future diagnostics programs, and we can eventually expect this to become readily accessible to teams across Europe as sequencing becomes more affordable. Our work sets the stage in preparedness for this event. We know that many European laboratories may not have access to metagenomics for pathogen identification. We feel that this and other approaches can serve as a platform for epidemic preparedness.

We are aware that long-read sequencing technologies are becoming affordable, accessible technologies for doing disease diagnostics. One of these platforms is from Oxford Nanopore Technologies (ONT). ONT allows for long-read sequencing of lengths over 5 kb, in contrast to short-read read sequencing, such as Illumina, which has a maximum of 300 to 500 bp. Long-read sequencing, compared to short-read, provides longer contiguous sequences, which is critical for important repetitive virulence factors or insertion elements. Some disadvantages of long-read compared to short-read sequencing include (i) fewer total reads generated and (ii) often higher inherent error rates due to the sequencing chemistry (34). We are concerned that fewer reads would not provide sufficient coverage to multiplex samples and would capture low concentrations of pathogen reads, while Illumina has more reads to reach a lower limit of detection. Higher error rates could affect classification methods for *Xf* subspecies, as their genomes display less than 3% difference for ANI. However, the technologies are swiftly advancing with new ONT chemistries (R10) that reduce the sequencing errors competitively with Illumina (34). Overall, a complete diagnostic approach should include long- and short-read sequencing to provide at the same time information about the genomes, pathogen abundance, and nucleotide changes.

In conclusion, our pipeline provides *Xf* taxonomy and functional information for diagnostics without extensive knowledge of the host or pathogen. The pipeline databases used for the analysis can be public repositories or privately collected gDNA and could be adapted by the user and tailored to different plant pathogens. The sequencing can be adapted to be an in-house system, as the library preparation and sequencing are user-friendly and not limited by the DNA quality or quantity. The analysis can be adjusted to detect several pathogens simultaneously. Our pipeline can be used for early detection of *Xf* or other crop pathogens and be incorporated as part of management strategies.

## MATERIALS AND METHODS

***Xylella fastidiosa* strains.** The strain *Xf* subsp. *fastidiosa* (*Xff*) CFBP 7970, isolated in the United States (Florida) in 2013 from *Vitis vinifera*, and *Xf* subsp. *multiplex* (*Xfm*) CFBP 8418, isolated in France in 2015 from *Spartium junceum*, were provided by the French Collection of Plant-Associated Bacteria (CIRM-CFBP; https://www6.inra.fr/cirm_eng/CFBP-Plant-Associated-Bacteria) and used as controls for whole-genome sequencing. Both strains were cultivated on modified PWG medium (Gelrite, 12 g · liter$^{-1}$; soytone, 4 g · liter$^{-1}$; Bacto tryptone, 1 g · liter$^{-1}$; MgSO$_4$·7H$_2$O, 0.4 g · liter$^{-1}$; K$_2$HPO$_4$, 1.2 g · liter$^{-1}$; KH$_2$PO$_4$, 1 g · liter$^{-1}$; hemin chloride [0.1% in NaOH, 0.05 M], 10 ml · liter$^{-1}$; bovine serum albumin [BSA] [7. 5%], 24 ml · liter$^{-1}$; L-glutamine, 4 g · liter$^{-1}$) at 28°C for 1 week.

**Artificially inoculated plant samples.** For artificial inoculations, 10 ml of a calibrated CFBP 7970 or CFBP 8418 strain suspension was spiked in 2 g of detached *V. vinifera* leaves. Sterile water was used for negative controls. The DNA extraction was performed using a CTAB-based extraction protocol (2) with slight modifications in order to concentrate bacterial DNA. After a 20-min centrifugation of the sample at 20,000 × *g*, the pellet was resuspended in 1 ml of CTAB buffer. At the end of the extraction protocol, the pellet was resuspended in 50 $\mu$l of sterile demineralized water. *Xf* presence in the infected samples was checked using Harper's qPCR assay (8).

**Plant material and bacterial gDNA.** Healthy plant material of *Vitis vinifera* (2 g of leaf petioles) was spiked with 10 ml of a calibrated CFBP 7970 or CFBP 8418 strain suspension. Sterile water was used as the negative control. The DNA extraction was performed using a CTAB-based extraction protocol (2) with slight modifications in order to concentrate bacterial DNA. After a 20-min of centrifugation at 20,000 × *g*, the plant macerate sample was resuspended in 1 ml of CTAB buffer. At the end of the extraction protocol, the pellet was resuspended in 50 $\mu$l of sterile demineralized water. *Xf* presence in the infected samples was checked using Harper's qPCR assay (8).

Naturally infected samples were collected in Europe and the United States. Symptomatic samples of *Olea europaea*, *Polygala myrtifolia*, and *Quercus ilex* were collected in October 2018 in Corsica (France) and in September 2019 in the French Riviera. *Xf* detection and DNA extraction of whole infected plant tissue were performed as mentioned above. *Xf* subspecies were identified using the tetraplex qPCR (17).

Twig tissues, leaf petioles, or green shoots of *Rhamnus alaternus*, *Spartium junceum*, and *P. myrtifolia* growing in the *Xf* outbreak zone of Monte Argentario (Grosseto, Tuscany, Italy) were collected during 2019 and 2020 (35). *Xf* was detected using Harper's qPCR assay (8). For samples with a $C_q$ value lower than 30, *Xf* isolation was attempted on buffered charcoal yeast extract (BCYE) agar according to PM 7/24-4 (2). Bacterial isolates

that became visible to the unaided eye within 3 days of incubation at 28°C were discarded; those that became visible thereafter were streaked twice for purity on BCYE agar and identified as *Xf* based on qPCR results (8). Reactions were carried out after boiling the bacterial suspension for 10 min. The DNA of one isolate among those that tested positive by qPCR from each plant was extracted using the CTAB-based protocol and further characterized to the subspecies and ST levels following the multilocus sequence typing (MLST) approach (36). The GoTaq probe qPCR master mix (Promega; A6102) and GoTaq G2 (Promega; M784B) polymerase were used for qPCR and conventional PCR experiments, respectively. The bacterial DNA of three isolates from *R. alaternus* (IT1-Ra to IT3-Ra), one from *S. junceum* (IT1-Sj), one from *P. myrtifolia* (IT1-Pm), and one from *Prunus dulcis* (IT1-Pd) were sequenced in this study.

*Vitis vinifera* DNA samples were received from the Virginia Tech Plant Disease Clinic (Virginia, USA). All samples were collected from Virginia vineyards in 2019. The US2-Vv sample was collected from a vineyard in Greene County, and the US3-Vv sample was from a vineyard in Isle of Wight County. The DNA extraction and *Xf* detection protocols are based on instructions from the Virginia Tech Plant Disease Clinic (VTPDC). Approximately 50 to 100 mg of grape leaf or petiole tissue was excised from each sample using a razor blade. The excised tissue was transferred to lysing matrix A tubes (MP Biomedicals; 6910-500) and ground using a FastPrep 24 device (MP Biomedicals; 116004500). DNA was extracted using an Isolate II plant DNA extraction kit (Bioline; 52070) the following manufacturer's recommendations and CTAB lysis buffer. For *Xf* detection, Harper's qPCR was performed on the StepOnePlus system (Life Technologies; 4376600) with a Sensi-FAST probe Hi-ROX qPCR kit (Bioline; 82005) (8).

**Pipeline controls.** To test the pipeline, seven samples were used as negative controls. The controls were two DNA samples from healthy barley and wild grass leaves that were grown in a greenhouse; two DNA samples from barley leaves were infiltrated with a bacterial suspension ($10^8$ CFU/ml) of *Xanthomonas translucens* pv. translucens UPB886. Three healthy samples were collected in France in 2020 and in the United States in 2019. Petioles and midribs were collected from healthy *Olea europaea* plants in a non-*Xf*-infected area (Angers, France) and from *Polygala myrtifolia* plants that were purchased form a local nursery. *V. vinifera* leaves were collected from the vineyard in Greene County, VA, USA. The DNA from these samples was extracted using the CTAB method as mentioned above. The absence of *Xf* in the healthy plants was confirmed using Harper's qPCR assay (8).

For *in silico* pipeline controls, two types of positive controls were used. To validate the detection of different concentrations of *Xf*, barley fasta sequence files were *in silico* mixed with reads of the *Xff* CFBP 7970 sequenced in this study. The final proportion of *Xff* reads in the sample ranged from 0.2 to 2.4% of total reads. To validate the detection limits for single and mixed infections, the barley fasta sequence files were *in silico* mixed with different proportions of *Xff* CFBP 7970 and *Xfm* CFBP 8418 reads, to get from percentage CFBP7970:99% CFBP8418 to 99% CFBP7970:1% CFBP8418.

**iSeq 100 sequencing.** iSeq 100 sequencing libraries were prepared according to the Illumina reference guide for the Nextera DNA Flex library prep and Nextera DNA CD indexes. In brief, 200 to 500 ng of DNA was quantified by spectrophotometry and used for library preparation. Then, the libraries were diluted to have the same starting concentration prior to sample pooling. Eight to twelve libraries were mixed together, and 1 nM pooled library was used per run. The sequencing settings were paired-ended (PE) read type, 151 read cycles, and 8 index cycles. In the iSeq 100 system, the Illumina Generate FASTQ analysis module for base calling and demultiplexing was selected. After sequencing for 17 h, fastq paired-end read files were extracted from the machine for subsequent analysis.

**Pipeline for *Xylella* sp. detection, classification, and quantification via metagenomic analysis.** Fastq files and the program Kraken 2 were used to recover *Xylella* reads (22). The Kraken 2 command options were –paired, –minimum-hit-groups 5, –report, and –db. The database was created with 92 NCBI genomes, 79 from *Xf*, 2 from *Xylella taiwanensis*, 10 from *Xanthomonas* spp., and 1 from *Escherichia coli* (Table S1). The 81 *Xylella* genomes were used to recover *Xylella* reads. The last 11 genomes were added to remove reads common to *Proteobacteria* that might give a false-positive match. The tool SendSketch with the nucleotide (nt) server was used to make sure the 81 NCBI *Xf* genomes were not contaminated with plant reads (BBMap; B. Bushnell; sourceforge.net/projects/bbmap/; 30 October 2019). A total of 49 NCBI sequences from plant 18S and chloroplast were added to the Kraken 2 customized database to avoid extracting reads annotated as plant reads (Table S1).

The reads classified as *Xylella* were extracted with the script extract_kraken_reads from the KrakenTools suite (https://github.com/jenniferlu717/KrakenTools). The extracted *Xf* reads were used for downstream analysis. First, *Xf* reads were *de novo* assembled with the software SPAdes (23) using default settings and the option –only-assembler. Second, the *Xf* contigs were the query sequences in the Basic Local Alignment Search Tool website (NCBI) to confirm if they were *Xf* reads or misclassified plant reads. The blastn parameters were nucleotide collection (nonredundant/nucleotide [nr/nt]) as database and megaBLAST program selection.

The *Xf*-positive contigs were used in four different analyses: (i) to identify subspecies, (ii) to reconstruct phylogeny, (iii) to identify the genetically closest strains already sequenced, and (iv) to identify alleles from specific genes and the MLST profile.

To identify subspecies, the tool SendSketch was run with the parameters, mode=sequence, records=2, format=3, minani=100, minhit=1, address=ref, and level=0 (BBMap; B. Bushnell B.; sourceforge.net/projects/bbmap/). Only contigs with 100% average nucleotide identity (ANI) were used to identify *Xf* subspecies. The *Xf* contigs with no hits were considered core sequences. For visualization, results were plotted using stacked bars.

To reconstruct phylogeny, ANI was calculated using the software Pyani (0.2.10) and LINbase (24, 25). For Pyani, the option -ANIm was set, and for LINbase, "Identify using a gene sequence" was set as the identification method. The R package ComplexHeatmap was used to visualize the ANI cluster analysis with the parameters clustering_distance_rows = robust_dist and clustering_method_rows = "average." Robust_dist was

a function suggested by the ComplexHeatmap Complete Reference (37). We confirmed the scale using hclust from the R library stats.

To identify the genetically closest already sequenced *Xf* strains, the tool BBSplit was run with *Xf*-contigs (BBMap; B. Bushnell; sourceforge.net/projects/bbmap/). The tool used 81 *Xf* genomes from NCBI and the options minratio=1 and ambig=best. For comparisons, each sample was normalized to its total *Xf* contigs and plotted using the R package ComplexHeatmap. For the isolated bacterial genomes, the most abundant strains were used as references to identify genomics variants using Harvest suite tools (26).

To determine specific gene alleles, the percentage of identity was calculated using *Xf* contigs as the query and the blastn algorithm (nucleotide-nucleotide BLAST 2.8.1+). The database contained complete nucleotide sequences for all alleles for the seven genes used for ST identification (*cysG*, *gltT*, *leuA*, *malF*, *nuoL*, *holC*, and *petC*) (36). All 147 alleles were downloaded from the website PubMLST (38) (last updated, 2019-03-06). To determine the presence of reported *Xf* virulence-related genes or those common to several plant-pathogenic bacteria (39, 40), the percentage of similarity was calculated using *Xf* contigs as the query using the blastx algorithm (nucleotide-nucleotide BLAST 2.8.1+). The databases contained complete amino acid sequences for gumBCDE, pilBMQTVW, rpfCEFG, tolC, 6-phosphogluconolactonase (pgl), and hemagglutinin from the *Xfm* M12 and *Xff* M23 NCBI genomes. For both blastn and blastx scripts the default settings were used, and two were modified (evalue 0.1; outfmt "6 qseqid sseqid pident evalue").

**MiSeq sequencing.** To validate the iSeq 100 results, the same iSeq 100 libraries were used for MiSeq deep sequencing. Nine samples were selected, at least one from each iSeq 100 run, making sure not to use the same i5 and i7 tags. These nine libraries were sent to the Animal Disease Diagnostic Laboratory (Ohio Department of Agriculture, Reynoldsburg, Ohio) for sequencing. Library preparation was performed using an Illumina DNA Flex kit, and 2 × 250 sequencing was performed on the MiSeq platform using V3 chemistry. The pipeline described above was used to analyze the MiSeq fastq files.

**Data availability.** The assembled genomes of the six *Xf* isolates were deposited in NCBI under the BioProject number PRJNA728043 and BioSample numbers SAMN19067399, SAMN19067400, SAMN19067398, SAMN19067401, SAMN19067403, and SAMN19067402. The scripts for Kraken 2 can be found at https://github .com/DerrickWood/kraken2/wiki/Manual; the scripts to analyze Kraken 2 outputs can be found at https:// github.com/jenniferlu717/KrakenTools; the scripts for SendSketch and BBSplit are from the user guide at https://jgi.doe.gov/data-and-tools/bbtools/bb-tools-user-guide/. To calculate the SNPs for each genome, we used the Harvest manual at https://harvest.readthedocs.io/en/latest/. For building the trees, we used the scripts from the ComplexHeatmap reference manual at https://jokergoo.github.io/ComplexHeatmap -reference/book/a-single-heatmap.html. All iSeq 100 and MiSeq raw reads can be made available upon request.

## SUPPLEMENTAL MATERIAL

Supplemental material is available online only.

**FIG S1**, EPS file, 2.4 MB.
**FIG S2**, EPS file, 0.9 MB.
**FIG S3**, EPS file, 0.7 MB.
**FIG S4**, EPS file, 1.1 MB.
**TABLE S1**, XLSX file, 0.02 MB.
**TABLE S2**, XLSX file, 0.03 MB.
**TABLE S3**, XLSX file, 0.01 MB.
**TABLE S4**, XLSX file, 0.02 MB.
**TABLE S5**, XLSX file, 0.02 MB.
**TABLE S6**, XLSX file, 0.01 MB.

## ACKNOWLEDGMENTS

We are grateful for funding support from the Ohio Department of Agriculture Specialty Crops block grant (AGR-SCG-19-03) and USDA NIFA FACT (2021-67021-34343) to J.M.J. Sara Campigli was financed by a grant from the Phytosanitary Service of the Tuscany Region (Italy).

We thank The Ohio Supercomputer Center for providing high performance computing resources.

We also thank Stephen Cohen, Jeff Chang, and Alexandra Weinsberg for their valuable input for the experiment development and editing.

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
