## [Reviewer comments · mSystems]

Metagenomic sequencing for identification of *Xylella fastidiosa* from leaf samples

Veronica Roman-reyna, Enora Dupas, Sophie Cesbron, Guido Marchi, Sara Campigli, Mary Ann Hansen, Elizabeth Bush, Melanie Prarat, Katherine Shiplett, Melanie Ivey, Joy Pierzynski, Sally Miller, Francesca Peduto Hand, Marie-Agnès Jacques, and Jonathan Jacobs

Corresponding Author(s): Jonathan Jacobs, Ohio State University

Review Timeline:

Submission Date:	May 12, 2021
Editorial Decision:	July 22, 2021
Revision Received:	July 27, 2021
Editorial Decision:	August 16, 2021
Revision Received:	August 18, 2021
Accepted:	September 8, 2021

Editor: Davide Bulgarelli

Reviewer(s): Disclosure of reviewer identity is with reference to reviewer comments included in decision letter(s). The following individuals involved in review of your submission have agreed to reveal their identity: Alessio Aprile (Reviewer #1)

Transaction Report:

DOI: <https://doi.org/10.1128/mSystems.00591-21>

July 22, 2021

Prof. Jonathan M Jacobs
Ohio State University
Columbus

Re: mSystems00591-21 (Metagenomic sequencing for rapid identification of *Xylella fastidiosa* from leaf samples)

Dear Prof. Jonathan M Jacobs:

Thank you for submitting your manuscript to mSystems. We have completed our review and I am pleased to inform you that, in principle, we expect to accept it for publication in mSystems. However, acceptance will not be final until you have adequately addressed the reviewer comments.

In addition, the maximum number of supplementary items per submission (figures and table) is 10, therefore please re-arrange your current material accordingly.

Finally, but equally important, a requisite for final acceptance is the presence of a 'Data Availability' paragraph where all the accession numbers of the generated sequences and scripts for data analysis can be retrieved (this paragraph is currently missing).

Preparing Revision Guidelines

For complete guidelines on revision requirements for your article type, please see the journal Article Types requirement at <https://journals.asm.org/journal/mSystems/article-types>. **Submissions of a paper that does not conform to mSystems guidelines will delay acceptance of your manuscript.**

Sincerely,

Davide Bulgarelli

Editor, mSystems

Journals Department
Reviewer comments:

Reviewer #1 (Comments for the Author):

The manuscript entitled "Metagenomic sequencing for rapid identification of *Xylella fastidiosa* from leaf 2 samples" reports a new strategy in the detection and contrast *Xylella fastidiosa*.

The common strategies to identify the bacterium involved ELISA and qPCR methods, but, since their weak limit of detection, they are ineffective in early disease identification and in the host eradication strategies.

The tool proposed by Roman-reyna et al., is very promising as demonstrated by experiments and bioinformatic analyses reported in this manuscript.

The main weakness of this strategy, in my opinion, is the lack of suitable equipment and specialized employees (bioinformaticians) in next-generation sequencing analyses at phytosanitary European institutes.

I suggest the following revisions:

INTRODUCTION:

1) Line 65: Reference 4 is not specific about tyloses. You can add, for example, Sabella et al., 2019 (Xylem cavitation susceptibility and refilling mechanisms in olive trees infected by *Xylella fastidiosa*)

2) Line 74-76. I partially disagree with the authors: The authors stated "fast and accurate detection" to prevent losses. Anyway, even with fast and accurate methods, we are not able to avoid the dispersion of the pathogen. The most important parameter in Xf detection (and host eradication) is the early detection. Methods with the lowest limit of detection should be developed. Since the method developed by the authors have a very low LOD, they should stress this point.

3) Line 87: please correct the typo "is".

4) Line 95: please remove the typo ";".

RESULTS

- 5) Line 129-132: I didn't understand very well. Can you explain better? Thanks.
- 6) Line 137: MLST. Please define it. The acronym is explained only on page 21.
- 7) Line 187-189: Figure S2. It is very difficult to distinguish the 0.081 on the y-axis. Why did you consider the ratio of 60:40 as mixed infection and then 80:20 as not mixed infection?
- 8) Line 287 to 298. Congratulation on this result. This is a key point for the future development of Xf detection tools.

Reviewer #2 (Comments for the Author):

Manuscript by Roman-Reyna et al., entitled 'Metagenomic sequencing for rapid...'

This study describes the set-up and use of a metagenomics pipeline using short reads via the iSeq in order to detect the plant pathogen *Xylella fastidiosa*. The detection system proved effective and very sensitive.

General comment

This pipeline is very powerful and allows sensitive detection and classification of *X. fastidiosa* from plant samples. It is described as an alternative to the current standard methodology of quantitative real time PCR (qPCR); its sensitivity and more in depth characterization provides clear advantages. It is however not a convenient, fast or cheap alternative since it requires NGS equipment and computational know-how. Detection methods of plant pathogens are advisable to be fast, cheap and requiring as little equipment and know as possible. This methodology has however many advantages resulting in a more in depth analysis of the plant samples. It can also be adapted to different pathogens or even pathobiome analysis.

Specific comments:

1. The analysis most likely resulted in the identification of many more microorganisms; authors have not presented or commented on this data. It would have been interesting to learn the significant presence of other microorganisms co-occurring with *X. fastidiosa*.
2. Explain more clearly in the discussion the advantages/disadvantages of using short reads.
3. Nanopore sequencing technology is evolving quickly and in the near future it will be a very cheap, quick and easily adaptable techniques. Possibly this technology can be better suited to pathogen detection since it is a more 'movable' field-like technology. It is encouraged to discuss this as a possible alternative NGS technology for this pipeline.
4. I recommend to remove the word 'rapid' from the title.

Please see below a Point-by-point responses to the issues raised by the reviewers.

Responses to the Editor:

We added a 'Data Availability' paragraph indicating the accessions number on NCBI. The scripts from this work are based on the default settings of each program manuals. We added the links to each manual website.

We re-arranged the supplementary items to be 10 figures and tables in total.

Reviewer #1 comments and author responses:

Comment: The manuscript entitled "Metagenomic sequencing for rapid identification of *Xylella fastidiosa* from leaf 2 samples" reports a new strategy in the detection and contrast *Xylella fastidiosa*. The common strategies to identify the bacterium involved ELISA and qPCR methods, but, since their weak limit of detection, they are ineffective in early disease identification and in the host eradication strategies. The tool proposed by Roman-reyna et al., is very promising as demonstrated by experiments and bioinformatic analyses reported in this manuscript. The main weakness of this strategy, in my opinion, is the lack of suitable equipment and specialized employees (bioinformaticians) in next-generation sequencing analyses at phytosanitary European institutes.

Response: We appreciate the concern about access to whole genome and metagenome sequencing for diagnostics at European institutes. We added a comment in the manuscript, please lines 426-431: "We hope metagenomics for pathogen diagnostics will serve as a model for future diagnostics programs and eventually expect this to become readily accessible to teams across Europe as sequencing becomes more affordable. Our work sets the stage in preparedness for this event. We know that many European laboratories may not have access to metagenomics for pathogen identification. We feel that this and other approaches can serve as a platform for epidemic preparedness."

Comment: Line 65: Reference 4 is not specific about tyloses. You can add, for example, Sabella et al., 2019 (Xylem cavitation susceptibility and refilling mechanisms in olive trees infected by *Xylella fastidiosa*)

Response: We appreciate the comment from this reviewer. We have updated the text to reflect this and will include the reference as suggested.

Comment: Line 74-76. I partially disagree with the authors: The authors stated "fast and accurate detection" to prevent losses. Anyway, even with fast and accurate methods, we are not able to avoid the dispersion of the pathogen. The most important parameter in Xf detection (and host eradication) is the early detection. Methods with the lowest limit of detection should be developed. Since the method developed by the authors have a very low LOD, they should stress this point.

Response: We appreciate the comment from this reviewer. We modified the paragraph to be: “The primary control strategy for diseases caused by Xf includes eradication of infected hosts via early detection. Therefore, developing methods with the lowest limit of detection are critical to prevent major losses for growers and future pathogen dispersal”.

Comment: 3) Line 87: please correct the typo "is".

Response: We have made this change.

Comment: 4) Line 95: please remove the typo ",,".

Response: We have made this change.

Comment: 5) Line 129-132: I didn't understand very well. Can you explain better? Thanks.

Response: We appreciate the comment from the reviewer and apologize for the confusion. Briefly many NCBI Xf genomes have plant genomic DNA contamination. We modified the paragraph (line 131-137) to clarify this as: “The custom-made database had plant sequences to avoid false positive results because we found some NCBI Xf genomes contained plant genomic DNA sequences. The plant DNA sequence hits had 100% identity to plant 18S or chloroplast reads. We could not remove all plant reads from the 81 NCBI Xf genomes. Therefore, the plant reads in the database serve as a filter to ensure plant reads were not misidentifying as Xf reads.”

Comment: 6) Line 137: MLST. Please define it. The acronym is explained only on page 21.

Response: We have defined this.

Comment: 7) Line 187-189: Figure S2. It is very difficult to distinguish the 0.081 on the y-axis. Why did you consider the ratio of 60:40 as mixed infection and then 80:20 as not mixed infection?

Response: We are grateful for this comment from the reviewer and apologize for the confusion. For the Figure S2, now Figure S1B-C, we added the lower log-ratio numbers to display the values lower such as 0.081. For the comment about the 80:20 ratio, we changed line 195 to: “Log-ratios below -0.012 are identified as single Xff”. We would like to thank the reviewer for finding and helping us clarify this confusion.

Reviewer #2 comments and author responses:

Comment: 1. The analysis most likely resulted in the identification of many more microorganisms; authors have not presented or commented on this data. It would have been interesting to learn the significant presence of other microorganisms co-occurring with *X. fastidiosa*.

Response: We are grateful for this reviewer presenting this excellent point. Ultimately an ideal study would include linking the pathogen to the greater microbiome. Our team has carried out similar analyses with rice leaves examining linkages between the microbiome and difference plant varieties (Roman-Reyna et al, 2020). We did not include follow-up microbiome analysis because the sequence read depth is lower for technologies like iSeq100. We have made a comment to reflect the reviewer's concern in the discussion as a potential future direction to build on this research. Please see lines: 407-414: "The low number of reads also hampers deep SNP diversity, intersubspecific homologous recombination analyses, and microbiome studies. Other sequencing systems, with higher reads output than the iSeq100, can also be used with this pipeline as the input is fastq files. With a high number of reads, the pipeline will provide better resolution to recover the MSLT genes and have enough sequencing depth to describe the microbial diversity in the sample. Complete diagnostic analysis should include microbiome diversity analysis to provide context of microbial interactions during disease."

Comment: 2. Explain more clearly in the discussion the advantages/disadvantages of using short reads. and

Comment: 3. Nanopore sequencing technology is evolving quickly and in the near future it will be a very cheap, quick and easily adaptable techniques. Possibly this technology can be better suited to pathogen detection since it is a more 'movable' field-like technology. It is encouraged to discuss this as a possible alternative NGS technology for this pipeline.

Response to comment 2 and 3: We are grateful for this reviewer presenting this point. We added a paragraph talking about Nanopore long-read sequencing and compared with short-read sequences. In lines 438-454 we added: "We are aware that long-read sequencing technologies are becoming affordable, accessible technologies for doing disease diagnostics. One of these platforms is from Oxford Nanopore Technologies (ONT). ONT allows for long-reads sequencing of lengths over 5Kb, in contrast to short-read read sequencing, like Illumina, which has a maximum of 300-500bp. Long-read sequencing, compare to short-read, provides longer contiguous sequences, which is critical for important repetitive virulence factors or insertion elements. Some disadvantages of long-read compared to short-read sequencing include 1) fewer total reads are generated and 2) often higher inherent error rates due to the sequencing chemistry (34). We are concerned that fewer reads would not provide sufficient coverage to multiplex samples and capture low concentrations of pathogen reads, while Illumina has more reads to reach a lower limit of detection. Higher error rates could affect classification methods for Xf subspecies as their genomes display less than 3% difference for ANI. However, the technologies are swiftly advancing with new ONT chemistries (R10) that reduce the sequencing errors competitively with Illumina (34). Overall, a complete diagnostic approach should include long- and short-read sequencing to provide at the same time information about the genomes, pathogen abundance and nucleotide changes."

Comment: 4. I recommend to remove the word 'rapid' from the title

Response: We removed the word rapid from the tittle.

August 16, 2021

Prof. Jonathan M Jacobs
Ohio State University
Columbus

Re: mSystems00591-21R1 (Metagenomic sequencing for identification of *Xylella fastidiosa* from leaf samples)

Dear Prof. Jonathan M Jacobs:

Thank you for submitting your manuscript to mSystems. I have completed my evaluation of your revision, which did not require an additional round of external comments, and I am pleased to inform you that, in principle, we expect to accept it for publication in mSystems. However, acceptance will not be final until you have adequately these last, very minor, points:

-Figure S2 does not seem to be cited in the main text. Shouldn't be "located" on or around paragraph 192-98, instead of Figure 2?

-Figure 3 file appears blurred, at least on my cpu: can you please upload a high resolution version of this figure?

-Figures 2 and 5A: what's the scale on those trees? Also, they did not seem bootstrapped: can you please comment on this?

-In a few instances you refer to "hope" regarding the application of the proposed method. I suggest a more "emotionally-detached" and result-based approach, something along the lines: this investigation (or our results) demonstrates that metagenomics can be efficiently integrated (or represent a valuable alternative) to conventional detection methods.

Else it is a well written manuscript proposing a timely and potential impactful use of metagenomics-congratulations!

Sincerely yours,

Davide Bulgarelli

Preparing Revision Guidelines

To submit your modified manuscript, log onto the eJP submission site at <https://msystems.msubmit.net/cgi-bin/main.plex>. Go to Author Tasks and click the appropriate manuscript title to begin the revision process. The information that you entered when you first submitted the paper will be displayed. Please update the information as necessary. Here are a few

examples of required updates that authors must address:

For complete guidelines on revision requirements for your article type, please see the journal Article Types requirement at <https://journals.asm.org/journal/mSystems/article-types>. **Submissions of a paper that does not conform to mSystems guidelines will delay acceptance of your manuscript.**

Sincerely,

Davide Bulgarelli

Editor, mSystems

Journals Department
Reviewer comments:

Please see below a Point-by-point responses to the issues raised by the Editor.

Responses to the Editor:

-Figure S2 does not seem to be cited in the main text. Shouldn't be "located" on or around paragraph 192-98, instead of Figure 2?

We thank the Editor for finding that typo. We added Fig. S2 on line 199.

-Figure 3 file appears blurred, at least on my cpu: can you please upload a high resolution version of this figure?

We uploaded a new file. To make sure is not blurry, we downloaded the file, and it has good resolution.

-Figures 2 and 5A: what's the scale on those trees? Also, they did not seem bootstrapped: can you please comment on this?

We thank the Editor for the comment. The trees did not use bootstrap. The distance is based on the algorithms propose by ComplexHeatmaps from the R package. We also confirm the scale using hclust. We added in line 598," We confirmed the scale using hclust from the R library stats".

-In a few instances you refer to "hope" regarding the application of the proposed method. I suggest a more "emotionally-detached" and result-based approach, something along the lines: this investigation (or our results) demonstrates that metagenomics can be efficiently integrated (or represent a valuable alternative) to conventional detection methods.

We thank the Editor for this comment. We changed the word hope in all the document. We changed or removed the word from the lines 41, 424, and 454.

September 8, 2021

Prof. Jonathan M Jacobs
Ohio State University
Columbus

Re: mSystems00591-21R2 (Metagenomic sequencing for identification of *Xylella fastidiosa* from leaf samples)

Dear Prof. Jonathan M Jacobs,

I have no further comment and I'd like to take this occasion to congratulate with all co-authors for the very interesting and timely manuscript.

Your manuscript has been accepted, and I am forwarding it to the ASM Journals Department for publication. For your reference, ASM Journals' address is given below. Before it can be scheduled for publication, your manuscript will be checked by the mSystems senior production editor, Ellie Ghatineh, to make sure that all elements meet the technical requirements for publication. She will contact you if anything needs to be revised before copyediting and production can begin. Otherwise, you will be notified when your proofs are ready to be viewed.

As an open-access publication, mSystems receives no financial support from paid subscriptions and depends on authors' prompt payment of publication fees as soon as their articles are accepted. =

Publication Fees:

We recognize that the video files can become quite large, and so to avoid quality loss ASM suggests sending the video file via <https://www.wetransfer.com/>. When you have a final version of the video and the still ready to share, please send it to Ellie Ghatineh at eghatineh@asmusa.org.

Sincerely,

Davide Bulgarelli
Editor, mSystems

Journals Department
Fig. S3: Accept
Fig. S2: Accept
Table S5: Accept
Table S3: Accept
Fig. S1: Accept
Table S6: Accept
Table S2: Accept
Fig. S4: Accept
Table S1: Accept
Table S4: Accept